# Antibody response to tetanus, diphtheria, poliomyelitis, hepatitis B, and *H. influenzae* b vaccines in allogeneic hematopoietic stem cell transplant adult recipients: A multicenter trial

Olivier Epaulard[1]*, Martin Carré[2], Eric Hermet[3], Violaine Corbin[4], Emmanuelle Tavernier[5], Elisabeth Botelho-Nevers[6], Etienne Daguindau[7], Anne-Sophie Brunel[8], Pierre-Simon Rohrlich[9], Karine Risso[10], Salomé Gallet[1], Nicolas Gonnet[11], Saber Touati[1], Marc Manceau[11], Anne Thiebault[2]

1 Université Grenoble Alpes, Infectiologie, CHU Grenoble Alpes, France, 2 Université Grenoble Alpes, Hématologie, CHU Grenoble Alpes, France, 3 Hématologie, CHU de Clermont-Ferrand, France, 4 Infectiologie, CHU de Clermont-Ferrand, France, 5 Hématologie, CHU de St Etienne, France, 6 Infectiologie, CHU de St Etienne, France, 7 Hématologie, CHU de Besançon, France, 8 Infectiologie, CHU de Besançon, France, 9 Hématologie, CHU de Nice, France, 10 Infectiologie, CHU de Nice, France, 11 Université Grenoble Alpes, CIC1406-INSERM, Université Grenoble Alpes, France

* oepaulard@chu-grenoble.fr

## Abstract

### Introduction

National and international guidelines recommend vaccinating hematopoietic stem cell transplant (HSCT) recipients, although relatively few studies have evaluated immunogenicity in adults. We therefore aimed to assess the immune response in adult allogeneic HSCT recipients vaccinated against tetanus, diphtheria, poliomyelitis, hepatitis B, and *H. influenzae* b.

### Method

We conducted a multicenter prospective study. HSCT recipients were included at least 6 months post-transplantation (maximum: 24 months) if blood CD19 + lymphocytes were ≥0.1 G/L and plasma immunoglobulin ≥ 4g/L, and if no immunosuppressive therapy was applied. They received the hexavalent pediatric combination vaccine for tetanus, diphtheria, poliomyelitis, hepatitis B, and *H. influenzae* b (and pertussis) at months 0, 1, 2, and 12 (in addition to other recommended vaccines). Plasma antibodies against the five valences were quantified at inclusion and 1 month after the third and fourth doses.

### Results

We included 104 HSCT recipients (median age: 58 years [IQR:48–64]). Study vaccination was initiated a median of 11 months [IQR:9–14] after transplantation. Median

**Data availability statement:** All relevant data for this study are publicly available from the OSF repository (https://doi.org/10.17605/OSF.IO/KYDU6).

**Funding:** This study was funded by the national French hospital research program (PHRC) (Grant N° PHRCI-16-016).

**Competing interests:** The authors have declared that no competing interests exist.

[IQR] values for CD19 and plasma gammaglobulin at inclusion were 0.3 [0.2–0.6] G/L and 7.9 [6.4–11.1] g/L, respectively. Seroprotection after three doses and after the M12 booster was achieved for 97.2% and 97.5% of participants for tetanus, 100% and 97.5% for diphtheria, 96.6% and 92.7% for poliomyelitis, 78.3% and 84.1% for hepatitis B, and 94.6% and 95.0% for *H. influenzae* b. Adverse effects were benign.

## Conclusion

Vaccination against these five infections initiated during the first year post-allograft is immunogenic and should be performed in every recipient not undergoing immuno-suppressive therapy.

## Trial registration

ClinicalTrials.gov NCT03402776

## Introduction

Allogeneic hematopoietic stem cell transplantation (HSCT) is a complex process that is mostly used for persons with hematologic malignancies, in particular acute leukemias, myelodysplastic syndromes, and non-Hodgkin lymphomas. Transplant infusion is preceded by high-dose chemotherapy (conditioning regimen): this intensification therapy, which allows for the maximal eradication of the malignant cell clone, is in parallel responsible for partial or complete myeloablation and the destruction of virtually all circulating leukocytes and their progenitors. Consequently, and even if plasma cell may be relatively less sensitive to conditioning [1], most of the immune memory acquired before transplantation is lost (including elimination of residual plasma cell by the allogeneic reaction), thus making necessary to perform in HSCT recipients *de novo* vaccinations post-transplant. Various recommendations for (re)immunizations against various infections and pathogens (diphtheria, tetanus, pertussis, poliomyelitis, *H. influenzae*, hepatitis B, pneumococcus, meningococcus, Covid-19, influenza, measles, mumps, rubella, varicella-zoster virus, and, if needed, human papillomaviruses and traveler's diseases) have been formulated at European [2] or North American [3] levels, and (among others) in France [4,5], Germany [6], the United Kingdom [7], and Australia [8].

However, these recommendations rely on studies performed mostly in pediatric subjects, particularly for vaccines given in the first months of life in the general population (e.g., diphtheria, tetanus, pertussis, poliomyelitis, and *H. influenzae*) [9–13], and less studies were performed in adults before 2020, apart from tetanus vaccine [14] and *H. influenzae* b vaccine [15]. We therefore aimed to determine the immune response to these vaccines in adult HSCT recipients and to identify the factors associated with a lack of immune protection.

## Methods

We conducted a multicentric prospective study in 5 French tertiary centers performing allogeneic HSCT.

## Inclusion profiles

The 2014 French guidelines [4] recommend immunizing all allogeneic HSCT recipients 3 months post-transplantation against pneumococcus (and recently Covid-19), 6 months post-transplantation against diphtheria, tetanus, pertussis, poliomyelitis, *H. influenzae* b, and hepatitis B, 12 months post-transplantation against meningococcus, and 24 months post-transplantation against measles, mumps, rubella, and varicella. Influenza immunization should be performed 6 months post-transplantation or 3 months if the influenza season is about to begin or ongoing. Apart from influenza, pneumococcus, and Covid-19, vaccination should be delayed in the case of immunosuppressive therapy. After detailing the study and the procedures associated with participation, and after obtaining written informed consent, we included adult allogeneic HSCT recipients at least 6 months after transplantation. In the case of immunosuppressive therapy (e.g., for severe graft-versus-host disease [GvHD]), inclusion was delayed until 3 months after its discontinuation, although the inclusion of subjects receiving ruxolitinib or photopheresis was not delayed. If patients were receiving polyclonal gamma-globulins, inclusion was delayed for 3 months after the last infusion, and participants were withdrawn from the study if they received polyconal gammaglobulins during study time. If a subject had CD19+ lymphocytes in blood under 0.1 G/L (i.e., the minimal value >0 in our center) and/or total gammaglobulin plasma concentration under 4 g/L, inclusion was delayed until correction. No patient was included more than 24 months post-transplantation. Subjects with active or past HBV infection (HBs antigen and/or anti-HBc antibody) were not included. The 1st patient was included on the 29th of May 2018, and the study was closed for inclusion on the 25th of November 2021.

## Immunization

To immunize participants, we used a four-dose schedule (months 0, 1, and 2 with a booster at month 12), as recommended by the French 2014 guidelines. The hexavalent vaccines used (pediatric combination vaccines, non-indicated in the adult general population) combined pertussis, *H. influenzae* b, diphtheria, tetanus, hepatitis B, and poliomyelitis antigens (Infanrix Hexa ® [GlaxoSmithKline, Rueil-Malmaison, France] or Hexyon® [Sanofi-Pasteur, Lyon, France]) with non-reduced amounts of diphtheria and pertussis antigen. After each vaccination, adverse effects were actively sought by a phone call in the days following the injection. The serum concentration of antibodies directed against tetanus and diphtheria toxins (Virotech™, Dietzenbach, Germany), *H. influenzae* b polysaccharide (Vacczyme™, the Binding Group, Birmingham, UK), hepatitis B surface antigen (Roche Diagnostics™, Meylan, France), and poliomyelitis virus 1 (Cerba™, Frépillon, France) was determined at inclusion (M0, before vaccination), 1 month after the third dose (M3), and 1 month after the fourth dose (M13). In case of delayed immunization, samples were drawn only after the appropriate number of doses were administered (e.g., if the third dose was administered 3 months after the second, the sampling was performed at least 1 month afterwards and not 3 months post-inclusion). Antibody concentrations were assessed in each different center, without centralization. The thresholds considered for seroprotection were 0.1 IU/mL for diphtheria, 0.01 IU/mL for tetanus, 10 mIU/mL for hepatitis B, 1 μg/mL for *H. influenzae* b, and 1/8 dilution titer or 0.5 UI/ml for poliovirus [16]. We did not collect if (and which) other vaccines (e.g., meningococcus, pneumococcus, influenza, or Covid-19) were performed at the same time as the study injections.

## Original study design

The primary objective of the study was to determine whether subjects with seroprotection against less than four of the five concerned infections after three doses would benefit from an additional fourth vaccine dose between M3 and M5: antibody levels were assessed at M3 (after 3 doses), and a randomization would apply to those not seroprotected against 4 or 4 infection to received or not (unblinded) this additional dose. The endpoint was the measurement of specific antibodies against the five pathogens at M13. However, less than 10% participants were protected against less than four infections after the third dose (and only 3 were randomized to receive an additional dose). The inclusions were therefore stopped for futility after 104 inclusions.

## Final study design

The study finally focused on the immune response in the participants included but not on the effect of an additional vaccine dose. We analyze:

- the antibody levels of all participants at inclusion and at M3 (1 month after the third dose)

- the antibody levels at M13 (after the fourth dose) of all participants except for the 3 participants randomized to receive an additional dose between M3 and M5.

Noteworthily, we did not analyze the antibody levels right before the fourth dose (M13).

Statistical analyses were performed using R software (v4.3.1), and figures were produced using ggplot2 and the tidyverse environment. Confidence intervals for the proportion of participants showing satisfactory or unsatisfactory responses rely on a chi-square approximation. Links between the antibody levels at M13 and the number of covariates was assessed using multivariate linear regression analyses. All analyses are considered as exploratory, and a significance level of 0.05 was thus retained without correction for the multiplicity of tests.

The study received ethical approval from the *Comité de protection des personnes Sud-Ouest et Outremer 1* (N° Eudract 2017-003523-30) on December 19, 2018. It was registered on the clinicaltrials.gov database as under the number NCT03402776.

The funder had no role in study design, data collection and analysis, decision to publish, or preparation of the manuscript.

## Results

### Population

A total of 104 HSCT recipients were included between May 2018 and November 2021 (Fig 1). Table 1 summarizes the patient characteristics. The majority of participants had received an HSCT for acute myeloid leukemia; the median time between the HSCT and study inclusion was 11 months. Twenty-one (20.2%) of the participants had a non-severe GvHD at the inclusion.

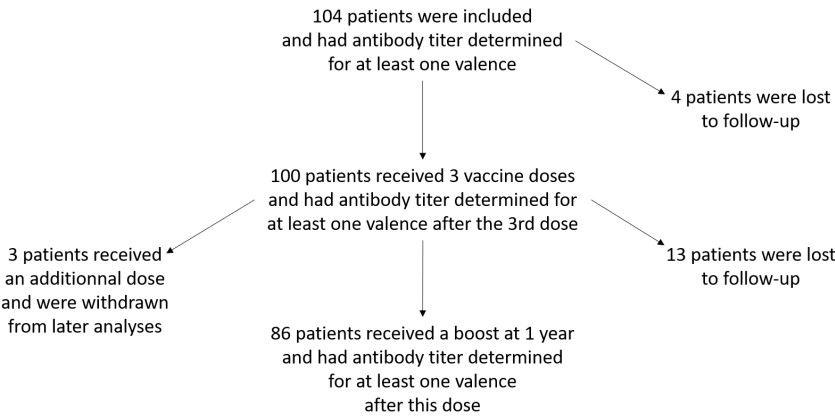

**Fig 1. Flow chart of the study.**

**Table 1. Patient characteristics.**

| Characteristics | | |
|---|---|---|
| Age* | | 58 [48-64] |
| Sex | Female | 39 (37.5%) |
| | Male | 65 (62.5%) |
| Reason for HSCT | acute lymphoid leukemia | 7 (6.7%) |
| | Acute myeloid leukemia | 67 (64.4%) |
| | Lymphoma | 7 (6.7%) |
| | Other | 23 (22.1%) |
| Conditioning regimen | Myeloablative | 36 (34.6%) |
| | Reduced intensity | 34 (32.7%) |
| | other | 34 (32.7%) |
| Time between HSCT and first hexavalent vaccine injection (months)* | | 11 [9 –14 ] |
| CD4 T cells at inclusion (G/L)* | | 0.3 [0.2-0.5] |
| CD8 T cells at inclusion (G/L)* | | 0.4 [0.2-1.4] |
| CD19 B cells at inclusion (G/L)* | | 0.3 [0.2-0.6] |
| Gammaglobulinemia (g/L)* | | 7.9 [6.4-11.1] |
| Time (months) between first hexavalent vaccine injection and booster (theoretically 12 months)* | | 13 [13 –14 ] Range: 12–37 |
| Time (days) between booster at month 12 and last antibody measure (theoretically 1 month)* | | 29 [34 –46] Range: 25–542 |

*(median [interquartile range]).

## Vaccine immunogenicity

Table 2 and Fig 2 show the antibody levels observed in participants. The study was designed to determine the extent to which an additional dose was useful in subjects with an unsatisfactory immune response after the first three doses. However, only six participants had a suboptimal immune response after three doses. Inclusions were therefore stopped after 104 inclusions, as the initial study size was designed for an incidence of suboptimal immune response of 50%. These six participants were randomized, with three of them receiving an additional dose; these three participants were therefore excluded from the analyses regarding the 1-year booster.

Table 3 shows the proportion of participants without seroprotection against at least one, two, three, four, or five pathogen(s). After the three initial doses or the 1-year boost, these proportions were very low.

For all vaccines, we observed that the antibody level at M13 did not significantly differ according to age, sex, type of hematological malignancy, delay between transplantation and vaccination, type of transplant conditioning, GvHD at inclusion, or (except when mentioned below) blood levels of gammaglobulins, CD4+T lymphocytes, or CD8+T lymphocytes at inclusion. The only association that emerged from multivariate linear regression linked the anti-diphtheria antibody level to CD8+T lymphocytes and the anti-*H. influenzae* b antibody level to both gammaglobulins and CD8+T lymphocytes.

## Vaccine safety

Table 4 shows vaccine safety. No major toxicity was observed after vaccination.

## Discussion

In this study, we aimed to assess the immune response to five vaccines currently recommended for adult recipients after allogeneic HSCT; to our knowledge, very few studies have explored responses to these vaccines in this population

**Table 2. Antibody response.**

| | | At inclusion | After three doses (M3) | After booster (fourth) dose at 1 year (M13)* |
|---|---|---|---|---|
| Diphtheria | participants analyzed | 98 | 94 | 81 |
| | Antibody level IU/mL median [interquartile range] | 0.26 [0.1-0.63] | 3.00 [1.54-3.00] | 3.70 [2.93-5.00] |
| | Antibody level IU/mL Geom. mean [95% CI] | 0.28 [0.22-0.35] | 2.24 [1.83-2.74] | 2.71 [2.19-3.35] |
| | Protected (≥ 0.1 IU/mL) | 25.3% | 100% | 97.5% |
| Tetanus | participants analyzed | 97 | 94 | 81 |
| | Antibody level IU/mL median [interquartile range] | 0.33 [0.12-0.68] | 1.76 [0.72-3.21] | 3.57 [1.56-5.00] |
| | Antibody level IU/mL Geom. mean [95% CI] | 0.32 [0.26-0.40] | 1.26 [0.98-1.63] | 2.05 [1.60-2.63] |
| | Protected (≥ 0.01 IU/mL) | 100% | 97.2% | 97.5% |
| Hepatitis B | participants analyzed | 98 | 88 | 68 |
| | Antibody level IU/L # median [interquartile range] | 3.1 [0-39.9] | 347 [22-1000] | 1000 [89-1000] |
| | Antibody level IU/L # Geom. mean [95% CI] | 7 [5 –11 ] | 110 [61-197] | 181 [101-352] |
| | Protected (≥ 10 IU/L) | 42.6% | 78.3% | 84.1% |
| *H. influenzae* | participants analyzed | 96 | 96 | 81 |
| | Antibody level 1 µg/mL ## median [interquartile range] | 0.985 [0.18-1.68] | 9 [9 –9 ] | 9 [9 –9 ] |
| | Antibody level 1 µg/mL ## Geom. mean [95% CI] | 0.67 [0.49-0.90] | 6.52 [5.54-7.68] | 6.54 [5.49-7.80] |
| | Protected (≥ 1 µg/mL) | 32.6% | 94.6% | 95.0% |
| Poliomyelitis | Patients analyzed | 99 | 99 | 82 |
| | Protected (≥ 0.5 UI/mL or dilution titer ≥ 1/8) | 71.6% | 96.6% | 92.7% |

* theoretically.

# values above 1 000 IU/L were given an arbitrary value of 1 001.

## values above 9 µg/mL were given an arbitrary value of 9.1.

(response to diphtheria, tetanus, polio vaccines: 4 studies [17–20], all but one smaller than ours; response to *Haemophilus influenzae*: 5 studies [17–21], all but one smaller than ours; response to hepatitis B: 2 studies [17,22], one being smaller than ours) in these subjects. We observed that the 3 + 1 vaccination schedule triggered protective humoral responses in a high proportion of participants: more than 95% had protective antibody concentration against all five pathogens. This is in line with the results obtained in previous studies (Conrad et al. [17]: more than 90% protection for tetanus, diphtheria, *H. influenzae* b infection, and 83% for hepatitis B; Sattler et al. [23]: 88–98% for tetanus, diphtheria, and *H. influenzae* b infection; Patel et al. [11]: 92–100% for tetanus, *H. influenzae* b, and poliovirus in children; Vance et al. [24]: more than 90% protection for tetanus and *H. influenzae* b infection). Hepatitis B had the lowest response rate at 84.1% after the 1-year booster, which was similarly observed in previous studies [17,22]. In our study, this lower anti-hepatitis B response may be related to the amount of vaccine antigen included in the hexavalent forms (10 µg instead of 20 µg usually used in adolescents and adults, and even 40 µg in, e.g., people with liver cirrhosis or receiving long-term dialysis). Overall, these results strongly support the vaccination of adult allogeneic HSCT recipients. However, it should be noted that the correlates of protection and the thresholds we used for seroprotection derived from studies in immunocompetent populations; it is unsure whether these correlates are identical in immunocompromised individuals such as HSCT recipients.

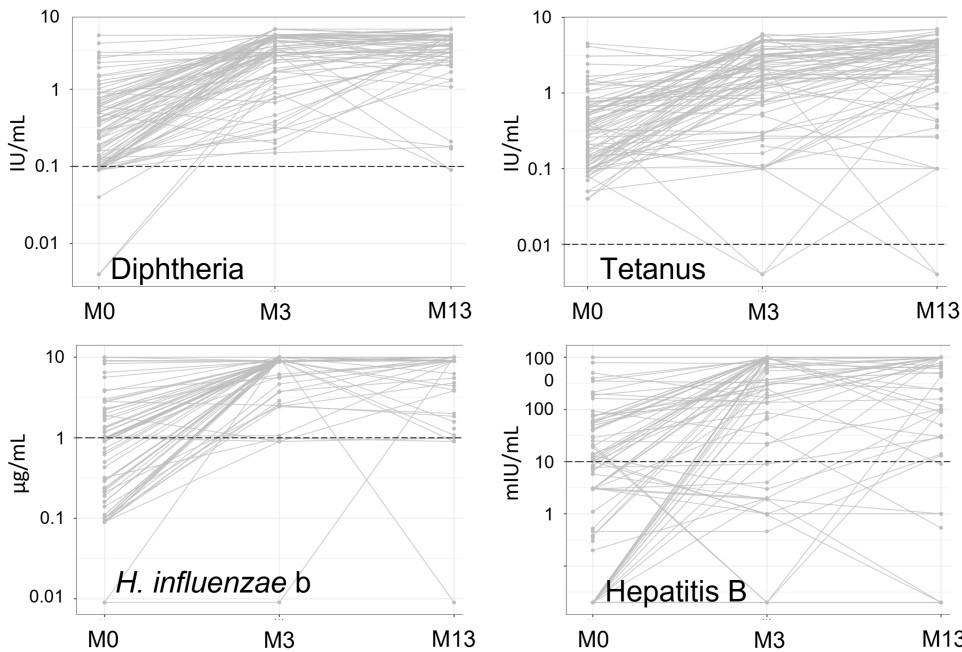

**Fig 2. Antibody levels before vaccination (M0), 1 month after the third dose (M3), and 1 month after the 1-year booster (M13).** The dashed line indicates the seroprotection threshold.

**Table 3. Proportion (%) of participants with no seroprotection.**

|  |  | At inclusion | After three doses (M3) | After booster (fourth) dose at 1 year* (M13) |
|---|---|---|---|---|
| Not protected against at least | 1 pathogen | 29.0% (20.3; 39.5) | 3.2 (0.8; 9.7) | 4.8 (1.5;12.4) |
|  | 2 pathogens | 0 (0; 4.9) | 2.1 (0.4; 8.2) | 2.4 (0.4; 9.1) |
|  | 3 pathogens | 0 (0; 4.9) | 2.1 (0.4; 8.2) | 2.4 (0.4; 9.1) |
|  | 4 pathogens | 0 (0; 4.9) | 2.1 (0.4; 8.2) | 2.4 (0.4; 9.1) |
|  | 5 pathogens | 0 (0; 4.9) | 0 (0; 4.9) | 1.2 (0.1; 7.4) |

* theoretically.

**Table 4. Adverse effects after each injection.**

|  | First injection (M0) | Second (M1) | Third (M2) | Fourth (M12) |
|---|---|---|---|---|
| Pain at injection site | 18.4% | 22.5% | 26.8% | 22.5% |
| Fever | 3.1% | 2.0% | 1.0% | 2.5% |
| Asthenia | 3.1% | 4.1% | 5.1% | 3.7% |

We did not observe any association between the delay from HSCT to the first vaccine dose (the median delay being less than 12 months in our study) and the antibody concentration after vaccination. This suggests that except in the case of immunosuppressive therapy, immunization with the five vaccines studied here can be initiated within 12 months post-transplantation. Of course, the decision should consider that later vaccination may be theoretically associated with a

better response, particularly when considering that (perhaps with the exception of *H. influenzae* b) the protection against these pathogens should be maintained lifelong. Interestingly, one study evidenced the persistence of responses for tetanus and diphtheria after a median of 14 years post-transplantation [9]. The case is nevertheless different for pathogens with a high risk of severe infection during the early months post-transplantation such as *S. pneumoniae*, SARS-CoV-2, and influenza virus for which it is relevant to initiate vaccination as early as 3 months post-HSCT.

Interestingly, a high proportion of participants had protective antibody levels even before vaccination, which was not associated with having received transplantation more recently. This residual immunity has long been observed, suggesting that even myeloablative regimens do not destroy all memory B cells [25]. These profiles may also suggest that memory B cells can be present in the donor graft. The impact of the donor immune status has already been evidenced for hepatitis B immunity [26,27], pneumococcus immunity [28], *H. influenzae* b and tetanus immunity [29,30], and potentially SARS-CoV-2 [31], although the data are not sufficiently robust to recommend donor vaccination. Moreover, all participants had received in the first 6 months post-vaccination 3 injections of a conjugated 13-valent vaccine with CRM-197 as a carrier protein; CRM-197 is a derivative of diphtheria toxin, and it has been shown that conjugated pneumococcal vaccine may boost anti-diphtheria immunity [32], and the antibody level we observed may have been increased by this conjugated vaccine.

Our study has several limitations. First, vaccine injections and antibody measures were not performed according to the planned schedule for a large proportion of participants, mainly because of the disorganization caused by the Covid-19 pandemic, leading the hospital to delay non-urgent consultations and non-urgent medical procedures such as vaccination and blood analyses. However, in this respect, our study reflects the routine shortcomings of post-transplantation follow-up, although the high proportion of participants with seroprotection is reassuring. Second, we did not assess some B lymphocyte subpopulations before or after inclusion. Indeed, the interplay between naïve and memory B lymphocytes [33] or between naïve and memory T lymphocytes [34], for example, is an important parameter of the immune system reconstitution post-transplantation, and it has been suggested that the vaccination schedule could be personalized in this regard [35]. Third, we chose a threshold of 0.01 IU/mL for tetanus seroprotection, although 0.1 IU/mL is more frequently used; this may have led us to overestimate the protection conferred by vaccination. However, after three doses, only one patient had antibodies against tetanus toxin between 0.01 and 0.1 IU/mL (0.09), and no patient had this profile after four doses; the choice of this threshold therefore had no or minimal impact on our conclusions. Fourth, the 2 vaccine combinations we used (Infanrix hexa®, GSK, and Hexyon®, Sanofi Pasteur) are no totally identical in antigen quantities of *Haemophilus influenzae* type b polysaccharide, of diphtheria toxoid, of inactivated poliomyelitis virus, although the obtained immune response in the infants is the same with these 2 vaccines; the variations in the use of these 2 vaccine combinations during the study, and the fact that both may have been administered in the same participant along the study, prevent the analysis of the impact of one form or another on the immune response. In addition, we did not assess the immune response to pertussis, which is currently reemerging; studies reported a lower immune response (68% seroprotection) for this pathogen after post-transplant vaccination [23], although the lack of clear correlates of protection for pertussis had led us not to assess the immunogenicity of this valence. Moreover, we did not analyze the impact of a past, pre-inclusion GvHD: therapy for GvHD may indeed have a long-lasting influence on future immune response. We also did not assess the influence of concurrent vaccinations, although some interference may occur between vaccines and modify their respective immunogenicity. Finally, as recommended in France when this study was designed, and to ensure a better long-term vaccine response, we did not include participants until they had a plasma gammaglobulin concentration ≥ 4G/L and blood CD19 + lymphocytes ≥0,1 G/L; this restriction is not recommended in most of guidelines nowadays.

## Conclusion

Vaccination of adult HSCT recipients against diphtheria, tetanus, *H. influenzae* b, poliomyelitis, and hepatitis B is safe and provides seroprotection in most subjects. The current recommendations to initiate vaccination for these pathogens or

diseases 6–9 months after transplantation are thus appropriate. Apart from hepatitis B, it may not be necessary to assess post-vaccine antibody levels.

## Author contributions

**Conceptualization:** Olivier Epaulard, nicolas gonnet, saber touati, anne thiébault.

**Data curation:** martin carre, eric hermet, violaine corbin, emmanuelle tavernier, elisabeth botelho-nevers, etienne daguindau, anne-sophie brunel, pierre-simon rohrlich, karine risso, salome gallet, saber touati, marc manceau.

**Formal analysis:** Olivier Epaulard, marc manceau.

**Funding acquisition:** Olivier Epaulard, anne thiébault.

**Investigation:** martin carre, eric hermet, violaine corbin, emmanuelle tavernier, elisabeth botelho-nevers, etienne daguindau, anne-sophie brunel, pierre-simon rohrlich, karine risso, salome gallet, saber touati.

**Methodology:** nicolas gonnet, saber touati.

**Project administration:** Olivier Epaulard, saber touati.

**Supervision:** Olivier Epaulard, saber touati.

**Validation:** Olivier Epaulard.

**Visualization:** Olivier Epaulard, marc manceau.

**Writing – original draft:** Olivier Epaulard.

**Writing – review & editing:** Olivier Epaulard, martin carre, eric hermet, violaine corbin, emmanuelle tavernier, elisabeth botelho-nevers, etienne daguindau, anne-sophie brunel, pierre-simon rohrlich, karine risso, salome gallet, nicolas gonnet, saber touati, anne thiébault.

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
