## [Decision Letter · Decision Letter 0]

15 Sep 2025

Dear Dr. Epaulard,

We look forward to receiving your revised manuscript.

Kind regards,

Ray Borrow, Ph.D., FRCPath

Academic Editor

PLOS ONE

Journal Requirements:

This study was funded by the national French hospital research program (PHRC) (Grant N° PHRCI-16-016).

5. Please note that funding information should not appear in any section or other areas of your manuscript. We will only publish funding information present in the Funding Statement section of the online submission form. Please remove any funding-related text from the manuscript.

Reviewers' comments:

Reviewer's Responses to Questions

**Comments to the Author**

1. Is the manuscript technically sound, and do the data support the conclusions?

Reviewer #1: Partly

Reviewer #2: Partly

Reviewer #3: Yes

2. Has the statistical analysis been performed appropriately and rigorously?

Reviewer #1: Yes

Reviewer #2: I Don't Know

Reviewer #3: Yes

3. Have the authors made all data underlying the findings in their manuscript fully available?

Reviewer #1: Yes

Reviewer #2: Yes

Reviewer #3: Yes

4. Is the manuscript presented in an intelligible fashion and written in standard English?

Reviewer #1: Yes

Reviewer #2: Yes

Reviewer #3: Yes

Reviewer #1: The manuscript #PONE-D-25-32490 titled “Antibody response to tetanus, diphtheria, poliomyelitis, hepatitis B, and H. influenzae b vaccines in allogeneic hematopoietic stem cell transplant adult recipients: A multicenter trial” described antibody response to tetanus, diphtheria, hepatitis B, polio and H. influenzae b after 4 doses of DTaP-HBV-Hib-IPV in adults after hematopoietic stem cell transplant. The study was interesting. I have the following comments,

1. In my opinion, a lot of studies have demonstrated good response of immune response after vaccination in patients after HSCT. The authors may have raise the novel issues this study contributed to.

2. In result part, the percentages of patients having protected antibody seemed to decrease in tetanus, diphtheria, polio. What would be the explanation of these results? In addition, why were the percentages of patients not protected for pathogens higher after the booster dose compared with after the 3rd dose? In fact, these percentages should be lower after the booster dose. Did some of these patients get additional immunosuppressive drugs during the vaccination course?

3. Did the presence of GvHD affect immune response to vaccination?

4. Concomitant vaccinations should be presented.

5. Minor grammatical correction should be performed.

Reviewer #2: Considering that the 2 vaccines used have different formulations (DTap2 vs DTap3) and that the immunogenicity values differ in both, should they have been analyzed independently in order to decide withether DTaP 3 is better or not (Hexavalent vaccines: What can we learn from head-to-head studies?

Markus Knuf , Hervé Haas, Pilar Garcia-Corbeira , Elisa Turriani , Piyali Mukherjee ,Winnie Janssens , Valérie Berlaimont Vaccine Volume 39, Issue 41, 1 October 2021, Pages 6025-6036)(Hexavalent CMIvaccines for immunization in paediatric age S. Esposito, C. Tagliabue, S. Bosis, V. Ierardi, M. Gambino and N. Principi Clinical Microbiology and Infection, Volume 20 Supplement 5, May 2014 76-85

Reviewer #3: Useful research and give good protection to the transplant patients . Excellent efforts .and choosing these type of research is really needed to the dither generation . Transplant patient survival and morbidity free survival is very important and for that this research is very useful .

**Do you want your identity to be public for this peer review?** For information about this choice, including consent withdrawal, please see our Privacy Policy

Reviewer #1: No

Reviewer #2: **Yes: ** Dr.M.D.Ravi

Reviewer #3: **Yes: ** Inumarthi Vara Padmavathi

---

## [Author Response · Author response to Decision Letter 1]

25 Sep 2025

Manuscript PONE-D-25-32490

Antibody response to tetanus, diphtheria, poliomyelitis, hepatitis B, and H. influenzae b vaccines in allogeneic hematopoietic stem cell transplant adult recipients: A multicenter trial

Response to editorial comments

We ensured this.

We created a totally de-identified set of data and deposit it on the OST (Center for Open Science) repository, with the DOI 10.17605/OSF.IO/KYDU6 on the OSF repository (osf.io).

We added this statement at the end of the manuscript.

We harmonized the way things were presented in the manuscript and on the submission website.

We ensured this.

This study was funded by the national French hospital research program (PHRC) (Grant N° PHRCI-16-016).

This statement is correct. We added this sentence in the revised version of the manuscript.

5. Please note that funding information should not appear in any section or other areas of your manuscript. We will only publish funding information present in the Funding Statement section of the online submission form. Please remove any funding-related text from the manuscript.

We removed the other funding-related text in the revised version of the manuscript.

We did not cite new articles in the revised version of the manuscript.

We performed this review. Particularly, no retracted articles were cited.

Response to reviewers' comments

Reviewer #1

The manuscript #PONE-D-25-32490 titled “Antibody response to tetanus, diphtheria, poliomyelitis, hepatitis B, and H. influenzae b vaccines in allogeneic hematopoietic stem cell transplant adult recipients: A multicenter trial” described antibody response to tetanus, diphtheria, hepatitis B, polio and H. influenzae b after 4 doses of DTaP-HBV-Hib-IPV in adults after hematopoietic stem cell transplant. The study was interesting. I have the following comments,

1. In my opinion, a lot of studies have demonstrated good response of immune response after vaccination in patients after HSCT. The authors may have raise the novel issues this study contributed to.

Only few studies assessed the antibody response to 5 pathogens/diseases vaccines in adult HSCT recipients (1 when this study was designed, and 3 when the manuscript was submitted to PLOS One). We added a sentence in the discussion to enlighten this novelty.

2. In result part, the percentages of patients having protected antibody seemed to decrease in tetanus, diphtheria, polio. What would be the explanation of these results?

The decrease is very low, or non-existent, between months 3 and 13 (tetanus: 97.2% and 97.5%, diphtheria: 100% and 97.5%, polio: 96.6% and 92.7%). It may be debatable to speculate on such a small difference.

In addition, why were the percentages of patients not protected for pathogens higher after the booster dose compared with after the 3rd dose? In fact, these percentages should be lower after the booster dose. Did some of these patients get additional immunosuppressive drugs during the vaccination course?

For this parameter too, the differences are very limited (and the proportion of non-protected are low, less than 5%), and may reflect the non-protection of a single one participant; we are maybe not sure that we should comment in the “Discussion” section such tiny variations.

3. Did the presence of GvHD affect immune response to vaccination?

As we excluded patients with a current GvHD, we did not have this data. However, we did not analyze the impact of a past GvHD. This has would indeed be relevant; it is a limitation, and we noted this in the “Discussion” section of the revised manuscript.

4. Concomitant vaccinations should be presented.

This indeed important. We do not have the data for each patient. For example, anti-meningococcal vaccination (B vaccine, conjugated polysaccharide tetravalent ACWY vaccine) is generally initiated alongside with the hexavalent vaccination; according to the season, influenza vaccine could be also co-administered; Covid-19 vaccine boost may have also been co-administered. We add this point in the revised version of the manuscript.

5. Minor grammatical correction should be performed.

Thank you for this remark. We performed these corrections.

Reviewer #2

Considering that the 2 vaccines used have different formulations (DTap2 vs DTap3) and that the immunogenicity values differ in both, should they have been analyzed independently in order to decide withether DTaP 3 is better or not (Hexavalent vaccines: What can we learn from head-to-head studies?

Markus Knuf , Hervé Haas, Pilar Garcia-Corbeira , Elisa Turriani , Piyali Mukherjee ,Winnie Janssens , Valérie Berlaimont Vaccine Volume 39, Issue 41, 1 October 2021, Pages 6025-6036)(Hexavalent CMIvaccines for immunization in paediatric age S. Esposito, C. Tagliabue, S. Bosis, V. Ierardi, M. Gambino and N. Principi Clinical Microbiology and Infection, Volume 20 Supplement 5, May 2014 76-85

The reviewer probably refers to names such as DT3aP-HBV-IPV/Hib used in the cited GSK article (we did no use tetravalent vaccines (DTap) but only hexavalent vaccines).

Indeed, several hexavalent vaccines were used during the study, as we did not restrict the study to one vaccine manufacturer (this would have led to increase too much the study costs, considering the funding received). This heterogeneity may have led to difference in vaccine immunogenicity, but in a way almost impossible to assess, as different vaccines may have been used in the same patient. However, this heterogeneity reflects the “real world” immunization practice, where hospital pharmacy may change from one year to another the manufacturer for a vaccine. We acknowledged this important point in the revised version of the manuscript (it was already acknowledged [“the 2 vaccine combinations we used (Infanric hexa® and Hexyon®) are no totally identical in antigen quantities of Haemophilus influenzae type b polysaccharide, of diphtheria toxoid, of inactivated poliomyelitis virus”], but we stressed more this point).

Reviewer #3

Useful research and give good protection to the transplant patients . Excellent efforts .and choosing these type of research is really needed to the dither generation . Transplant patient survival and morbidity free survival is very important and for that this research is very useful .

Thank you for this comment.

---

## [Decision Letter · Decision Letter 1]

29 Sep 2025

Dear Dr. Epaulard,

Thank you for submitting your manuscript to PLOS ONE. After careful consideration, we feel that it has merit but does not fully meet PLOS ONE’s publication criteria as it currently stands. Therefore, we invite you to submit a revised version of the manuscript that addresses the points raised below during the review process.

We look forward to receiving your revised manuscript.

Kind regards,

Ray Borrow, Ph.D., FRCPath

Academic Editor

PLOS ONE

Journal Requirements:

Reviewers' comments:

Reviewer's Responses to Questions

**Comments to the Author**

Reviewer #1: (No Response)

Reviewer #2: All comments have been addressed

2. Is the manuscript technically sound, and do the data support the conclusions?

Reviewer #1: Yes

Reviewer #2: Yes

3. Has the statistical analysis been performed appropriately and rigorously?

Reviewer #1: Yes

Reviewer #2: I Don't Know

4. Have the authors made all data underlying the findings in their manuscript fully available?

Reviewer #1: Yes

Reviewer #2: Yes

5. Is the manuscript presented in an intelligible fashion and written in standard English?

Reviewer #1: Yes

Reviewer #2: Yes

Reviewer #1: The manuscript #PONE-D-25-32490R1 titled “Antibody response to tetanus, diphtheria, poliomyelitis, hepatitis B, and H. influenzae b vaccines in allogeneic hematopoietic stem cell transplant adult recipients: A multicenter trial” described antibody response to tetanus, diphtheria, hepatitis B, polio and H. influenzae b after 4 doses of DTaP-HBV-Hib-IPV in adults after hematopoietic stem cell transplant. In my opinion, the previous issues were not quite well addressed.

1. In my opinion, no novel issues were pointed out from the existing knowledge. At least, please consider this reviewed article; doi: 10.3390/cancers13236140.

2. In result part, the percentages of patients having protected antibody seemed to decrease in tetanus, diphtheria, polio. What would be the explanation of these results? This should be better addressed.

3. Did the presence of GvHD affect immune response to vaccination? Previous GvHD could affect later immune response as immune reconstitution may be delayed. Addressing only CD19+ level and immunoglobulin level are not enough. T cell subset numbers or functions are also important in immune response.

4. Concomitant vaccinations should be presented. Interference of other vaccines may affect immune response. This should be clearly presented.

Reviewer #2: The terms DTaP 2 and DTaP 3 referred to the number of acellular pertussis components in the hexavalent vaccine. A large number of vaccine studies have used correlates of immunity aaginst pertussis - also accepted by vaccine regulatory bodies. Considering that it is re-emerging in a big way, I think this should have been looked at

**Do you want your identity to be public for this peer review?** For information about this choice, including consent withdrawal, please see our Privacy Policy

Reviewer #1: No

Reviewer #2: **Yes: ** Dr M.D Ravi

---

## [Author Response · Author response to Decision Letter 2]

7 Oct 2025

PONE-D-25-32490R1

Antibody response to tetanus, diphtheria, poliomyelitis, hepatitis B, and H. influenzae b vaccines in allogeneic hematopoietic stem cell transplant adult recipients: A multicenter trial

The funder had no role in study design, data collection and analysis, decision to publish, or preparation of the manuscript.

Responses to Review Comments to the Author (Revision N°2)

Reviewer #1

The manuscript #PONE-D-25-32490R1 titled “Antibody response to tetanus, diphtheria, poliomyelitis, hepatitis B, and H. influenzae b vaccines in allogeneic hematopoietic stem cell transplant adult recipients: A multicenter trial” described antibody response to tetanus, diphtheria, hepatitis B, polio and H. influenzae b after 4 doses of DTaP-HBV-Hib-IPV in adults after hematopoietic stem cell transplant. In my opinion, the previous issues were not quite well addressed.

1. In my opinion, no novel issues were pointed out from the existing knowledge. At least, please consider this reviewed article; doi: 10.3390/cancers13236140.

This comment is a bit surprising, as we are currently at the “revision” part of the process; meanwhile, this remark is understandable in the “initial submission” part of the process, not in the revision version (indeed, the data we reported in the revised version are not more or less “novel” than in the initial submission of the manuscript).

Moreover, as mentioned in the manuscript, this clinical trial brings immunogenicity data regarding the vaccine response in a rarely studied population: adult HSCT recipients. Only few studies reported such data (response to diphtheria, tetanus, polio vaccines: 4 studies, all smaller than ours; response to Haemophilus: 5 studies, all smaller than ours; response to hepatitis B in non-infected subjects: 2 studies, one being smaller than ours) in these subjects.

However, we modified again the R2 version of the manuscript to make this clearer, and added citations concerning small (N<30) studies.

Regarding the suggesting citation: the title is very interesting and promising, but we usually do not cite predatory editors, and therefore we prefer not to cite manuscripts published by MDPI journals (https://www.predatoryjournals.org/news/is-mdpi-predatory;
https://en.wikipedia.org/wiki/MDPI#Controversies), as giving publicity to such problematic stakeholders may be considered malpractice.

2. In result part, the percentages of patients having protected antibody seemed to decrease in tetanus, diphtheria, polio. What would be the explanation of these results? This should be better addressed.

We had stated in the “response to reviewers” document during the R1 revision process that “The decrease is very small, or non-existent, between months 3 and 13 (tetanus: 97.2% to 97.5%, diphtheria: 100% to 97.5%, polio: 96.6% to 92.7%). It may be debatable to speculate on such a small difference”.

3. Did the presence of GvHD affect immune response to vaccination? Previous GvHD could affect later immune response as immune reconstitution may be delayed.

This is indeed an important remark. We had already stated in the “response to reviewers” document that “As we excluded patients with a current GvHD, we did not have this data. However, we did not analyze the impact of a past GvHD. This has would indeed be relevant; it is a limitation, and we noted this in the “Discussion” section of the revised manuscript”.

On the other hand, we previously did not exploit the data available to us concerning the presence of non-severe GvHD at inclusion. We therefore did these analyses in the recent days and did not observe any difference between those with or without GvHD at inclusion. We added this result in the revised (R2) manuscript.

Addressing only CD19+ level and immunoglobulin level are not enough. T cell subset numbers or functions are also important in immune response.

This is an important remark, and indeed, we also measured TCD3, 4 and 8 lymphocytes populations in the participants, alongside with CD19. As stated since the initial submission of the manuscript, “For all vaccines, we observed that the antibody level at M13 did not significantly differ according to age, sex, type of hematological malignancy, delay between transplantation and vaccination, type of transplant conditioning, or (except when mentioned below) blood levels of gammaglobulins, CD4+ T lymphocytes, or CD8+ T lymphocytes at inclusion. The only association that emerged from multivariate linear regression linked the anti-diphtheria antibody level to CD8+ T lymphocytes and the anti-H. influenzae b antibody level to both gammaglobulins and CD8+ T lymphocytes”.

4. Concomitant vaccinations should be presented. Interference of other vaccines may affect immune response. This should be clearly presented.

We had stated in the “response to reviewers” document during the R1 revision process that “This is indeed important. [However, ] we do not have the data for each patient. For example, anti-meningococcal vaccination (B vaccine, conjugated polysaccharide tetravalent ACWY vaccine) is generally initiated alongside with the hexavalent vaccination; according to the season, influenza vaccine could be also co-administered; Covid-19 vaccine boost may have also been co-administered. We add this point in the revised version of the manuscript.”

We still thing that this comment is very relevant, but we still do not have the data. We have let in the revised (R2) version of the manuscript the sentence we had already introduced in the R1 revised version to acknowledge this (“We did not collect if (and which) other vaccines (e.g., meningococcus, pneumococcus, influenza, or Covid-19) were performed at the same time as the study injections”) and we added a sentence in the discussion section of the revised (R2) version of the manuscript.

Reviewer #2

The terms DTaP 2 and DTaP 3 referred to the number of acellular pertussis components in the hexavalent vaccine. A large number of vaccine studies have used correlates of immunity aaginst pertussis - also accepted by vaccine regulatory bodies. Considering that it is re-emerging in a big way, I think this should have been looked at.

This remark is very relevant: the issue of pertussis re-emergence is indeed concerning. Alas, when we designed this study, we decided not to analyze anti-pertussis antibody response. We may regret this, but we cannot produce analyses from data we purposely did not collect.

On this matter, we have let in the revised (R2) version of the manuscript the sentence we had already introduced in the R1 revised version to acknowledge this (“In addition, we did not assess the immune response to pertussis, which is currently reemerging; studies reported a lower immune response (68% seroprotection) for this pathogen after post-transplant vaccination [ref. 23], although the lack of clear correlates of protection for pertussis had led us not to assess the immunogenicity of this valence”).

---

## [Editor Report · Decision Letter 2]

8 Oct 2025

Antibody response to tetanus, diphtheria, poliomyelitis, hepatitis B, and H. influenzae b vaccines in allogeneic hematopoietic stem cell transplant adult recipients: A multicenter trial

PONE-D-25-32490R2

Dear Dr. Epaulard,

We’re pleased to inform you that your manuscript has been judged scientifically suitable for publication and will be formally accepted for publication once it meets all outstanding technical requirements.

Kind regards,

Ray Borrow, Ph.D., FRCPath

Academic Editor

PLOS ONE
---

## [Editor Report · Acceptance letter]

PONE-D-25-32490R2

PLOS ONE

Dear Dr. Epaulard,

I'm pleased to inform you that your manuscript has been deemed suitable for publication in PLOS ONE. Congratulations! Your manuscript is now being handed over to our production team.

Kind regards,

on behalf of

Prof. Ray Borrow

Academic Editor

PLOS ONE